# Design of Polysaccharide-Based Nanocomposites for Eco-Friendly Flexible Electronics

**DOI:** 10.3390/polym17121612

**Published:** 2025-06-10

**Authors:** Gabriela Turcanu, Iuliana Stoica, Raluca Marinica Albu, Cristian-Dragos Varganici, Mihaela Iuliana Avadanei, Andreea Irina Barzic, Lavinia-Petronela Curecheriu, Paola Stagnaro, Maria Teresa Buscaglia

**Affiliations:** 1Faculty of Physics, “Alexandru Ioan Cuza” University, Blv. Carol I, nr.11, 700506 Iasi, Romania; gabriela.irina@yahoo.com (G.T.); 2“Petru Poni” Institute of Macromolecular Chemistry, 41a Grigore Ghica Voda Alley, 700487 Iasi, Romania; stoica_iuliana@icmpp.ro (I.S.); albu.raluca@icmpp.ro (R.M.A.); varganici.cristian@icmpp.ro (C.-D.V.); mavadanei@icmpp.ro (M.I.A.); 3CNR-SCITEC, Institute of Chemical Sciences and Technologies “Giulio Natta”, National Research Council, 16149 Genoa, Italy; paola.stagnaro@cnr.it; 4CNR-ICMATE, Institute of Condensed Matter Chemistry and Technologies for Energy, National Research Council, 16149 Genoa, Italy

**Keywords:** biopolymer, ceramic nanofiller, morphology, refractivity, thermal properties, dielectric behavior

## Abstract

Flexible electronics is an applicative field in continuous expansion. This article addresses the requirements of this domain regarding eco-friendly and flexible components from a renewable chitosan polysaccharide that is progressively reinforced with barium titanate nanoparticles. Ultrafine ceramic powder was produced by the coprecipitation method, and the resulting phase composition and morphology were investigated by X-ray diffraction and transmission electron microscopy, together with the perovskite structure of the spherical nanoparticles. FTIR studies were conducted to elucidate the interactions between the two constituting phases of the composites. The filler dispersion in the matrix was checked by scanning electron microscopy. The rheological percolation threshold was compared with that extracted from electrical measurements. The thermal behavior was assessed by thermogravimetry and differential scanning calorimetry. The dielectric properties as a function of frequency and applied electric field were investigated, and the results are discussed in terms of extrinsic contributions. The current results demonstrate a straightforward method for producing tunable flexible structures.

## 1. Introduction

The challenges of our days regarding the advancement of electronic devices are cost reduction, elimination of toxic elements from components, development of eco-friendly processing techniques, dimensionality reduction, and obtaining flexible components primarily for medical and microelectronic fields [1,2,3,4]. Proposals for lead-free inorganic parts and the elaboration of eco-compatible procedures for producing inorganic materials have emerged to address at least some of these needs. These include the hydrothermal and precipitation methods that use lower processing temperatures and accessible raw materials. Particular attention was ascribed to the role of the dimensions in tailoring the properties of the ferroelectric and ferri/ferromagnetic materials and to find cost-effective and ecologically sustainable ways to densify bulk materials (e.g., cold sintering [5]).

The progress in flexible electronics has its origins in finding novel materials, including polymer-based semiconductors [6] and conductors [7], hydrogels [8], and sustainable natural bio-based materials [9]. Based on these, advanced and ecological components can be fabricated for subsequent integration into devices that combine flexibility and advanced operability [10,11]. Among the non-toxic polymers studied in recent years, chitosan has received huge importance due to its high abundance in nature and its great combination of properties, such as biodegradability [12], hydrophilicity [13], transparency in visible range [14], and mechanical resistance [15]. Moreover, this biopolymer has the advantage that it can be processed into various shapes, like beads, powders, porous membranes, films, flakes, or gels [16]. Intrinsically, chitosan is an electrical insulator and displays relatively low permittivity.

Hence, chitosan can be combined with other materials for manufacturing flexible and eco-friendly components, which can be incorporated into a variety of devices as a function of their physical properties. In view of making flexible substrates and active components, chitosan was blended with other macromolecular compounds, such as polyvinylpyrrolidone for higher thermal stability and flexibility [17], polyvinyl alcohol mixed with seed-based extract for modulating the optical and thermal features [18], and poly(trimethylene carbonate) for upgrading the mechanical properties [19]. For tailoring electrical resistivity, various types of materials are introduced into chitosan. For instance, gelatine and silver were incorporated in this biopolymer to attain conductive hydrogels for switching memory and transistors [20,21] or to make aerogels for selective adsorption [22]. Conductive hydrogels derived from chitosan have demonstrated excellent electrical performance and flexibility, which are critical for applications in wearable technology and artificial intelligence systems [20,21]. Furthermore, adjusting the dielectric performance, chitosan is mixed with other polymers, like poly(2-ethyl-2-oxazoline) [23], natural rubber [24], nitrile-modified cellulose nanocrystals [25], and methylcellulose doped with ammonium nitrate [26]. Another largely employed route for raising the permittivity of chitosan consists of the fabrication of composites containing ceramic fillers, namely, barium titanate [27], bismuth titanate [28], strontium titanate [29], aluminum oxide [30], or boron nitride [31]. The dielectric characteristics of polymer/ceramic composites are affected by the filler’s specific properties (shape, dimensions, polarizability), level of dispersion, orientation, amount inserted in the matrix, and interfacial compatibility with the polymer [32,33,34]. An important issue here is that the fraction of the ceramic phase from the polymer can determine relevant changes in the microstructure of the composite due to the percolation phenomenon. The steep jump in the electrical parameters is caused by the occurrence of path-like particle structures that allow charge carriers to flow [35]. Such non-linear changes in the electrical properties are also reflected in the shear flow behavior, so rheology is an alternative method to electrical one for elucidating the formation of filler network in composites [35,36,37,38]. Shear oscillatory tests are particularly sensitive to variations in the microstructure of the polymer nanocomposites [39]. The viscoelastic properties reflect the filler dispersion state and the interactions among the phases, especially when the percolated network is formed and a liquid-to-solid transition is taking place [40]. Therefore, there is an interrelationship between the percolation threshold (PT) extracted from electrical analysis and the rheological one. Most reports state that the electrical PT is higher than the rheological PT [41]. Such aspects are expected to influence the electrical and thermal behavior of the polymer nanocomposites. It was found that polymer filling with inorganic particles produces a stabilization effect on the thermal degradation of the macromolecular matrix, leading to enhanced permittivity [36,42].

Chitosan’s compatibility with barium titanate (BT) is significant in the context of developing piezoelectric materials. Chitosan/BT composites have shown improved dielectric and piezoelectric properties, which are essential for sensors and actuators in flexible electronics [43]. The piezoelectric effect can be exploited in pressure sensors and energy harvesting applications, allowing mechanical stress to generate electrical power; this could play a vital role in powering wearable devices sustainably [44]. The controlled integration of BT also allows the tuning of the electrical properties of flexible electronics, which can enhance device performance. In this context, this study sheds light on novel aspects that derive from extensive and deep characterization of the eco-friendly dielectrics for flexible electronic applications. This article describes the preparation of chitosan/ceramic BT and a technique for producing a large quantity of ultrafine barium titanate powders at ambient temperature. The small size of the obtained nanoparticles generates many interfaces with the biopolymer; hence, their implications on microstructure evolution and the composite performance are also monitored. The filler size and dispersion were studied by scanning electron microscopy. This was conducted by examining the interrelationship between the electrical and rheological PT and how this affects the material’s thermal stability. Furthermore, the dielectric properties are investigated by both electrical and non-electrical techniques to detect the accurate influence of the nanofillers on the chitosan’s permittivity. The high field electrical properties for chitosan/BT have rarely been documented, and the few reported studies mainly focus on the piezoelectric properties. According to our knowledge, such broad correlations on chitosan/BT systems have not been reported yet, and this aspect opens new paths towards profound understanding of dielectric materials designed for flexible electronics.

## 2. Materials and Methods

### 2.1. Materials

All the raw materials were purchased from Sigma-Aldrich, St. Louis, MO, USA: chitosan with medium molecular weight (>75% deacetylation); TiCl_4_ (99.9% purity); $Ba{Cl}_{2}\cdot 2H_{2}O$ (99% purity); NaOH (pellets, >95%); and glacial acetic acid (99.7% purity).

### 2.2. Experimental Details

The phase composition of BT powders was determined by X-ray diffraction (XRD) performed with a SHIMADZU XRD 6000, Kyoto, Japan, diffractometer with Ni-filtered CuKα radiation (λ = 1.5418 Å), a scan step of 0.02°, and a counting time of 10 s/step, for 2θ ranging between 10° and 80°.

The morphology and size of BT particles was checked by a transmission electron microscope (TEM) model HITACHI–HT7700 (Hitachi High Technologies Corporation, Tokyo, Japan). Morphology of the matrix and composite films was inspected in cross-section with a scanning electron microscope (SEM), Verios G4 UC (Thermo Scientific, Brno, Czech Republic).

Shear oscillatory properties of the matrix and composite solutions were investigated on a Bohlin CS50 instrument (Malvern Instruments, Malvern, UK). UV-VIS characteristics of the film samples were registered on a SPECORD 210 PLUS instrument (Analytik Jena GmbH, Jena, Germany).

Thermogravimetric analysis (TGA) measurements were conducted on a STA 449F1 Jupiter device (Netzsch, Selb, Germany). Around 11 mg of each sample was heated in alumina crucible at a 10 °C/min heating rate and in nitrogen atmosphere (flow rate 50 mL/min).

Differential scanning calorimetry (DSC) curves were recorded on a 200 F3 Maia device (Netzsch, Selb, Germany). Around 6 mg of each sample was heated in aluminum crucibles with pressed, pierced, and sealed shut lids. A heating rate of 10 °C/min was applied. The experiments were conducted in nitrogen atmosphere (flow rate 50 mL/min).

The infrared measurements were carried out with a Bruker Vertex-70 spectrometer (Bruker Optics, Ettlingen, Germany) equipped with a Golden Gate^TM^ ATR device. The wavenumber domain was 4000–600 cm^−1^, and the spectra were recorded at 2 cm^−1^ resolution as an average of 128 scans. The bands of interest were spectrally decomposed by curve-fitting with Voigt functions. The data manipulation was made in the OPUS 6.5 software.

Refractometry analyses were performed on the Abbemat MW apparatus (Anton Paar GmbH, Ashland, VA, USA). Broadband dielectric spectroscopy was carried out at room temperature using Solartron 1260 (Solartron Analytical, Hampshire, UK) for frequencies ranging from 1 Hz to 1 MHz. To determine the high-field properties, the selected composite films were immersed in silicon oil bath and subjected to voltage up to 5 kV. Subsequently, they were measured using the Solartron 1260 system described above.

### 2.3. Synthesis of Inorganic Filler and Composites Preparations

Ultrafine BaTiO_3_ (BT) powders were prepared by coprecipitation method, following a modified procedure described in the literature [45]. Briefly, aqueous solutions of $Ti{Cl}_{4}$ and $Ba{Cl}_{2}\cdot 2H_{2}O\left(Aldrich,99.9\%\right)$were quickly mixed with NaOH solution, followed by subsequent heating at temperatures exceeding 80 °C [46]. Considering previous studies [46], the [Ba]/[Ti] ratio was 1.04, and the concentration of the NaOH solution was required to have [OH^−^] = 1 mol/dm^3^ after precipitation of the BT. The concentration of the NaOH solution allows maintaining the pH = 14 during precipitation. Afterwards, the suspension was washed with distilled water until the pH became neutral. Finally, the obtained wet powder was frozen and lyophilized for 24 h.

To prepare the composites films, the BT nanopowders were dispersed in 1.5 wt.% chitosan solution by magnetic stirring for 3 h and then ultrasonicated at 40 kHz for 30 min. Finally, 25 mL of each nanocomposite solution (0.25; 0.5; 0.75, 1; 2.5, 5, and 10 vol%—corresponding to a mass ratio 3–56%) was placed in separate Petri dishes and dried 24 h at 50 °C to obtain films. The composites were named after BT content, from 0 for pure chitosan film to 10 for 10% BT volume content.

## 3. Results and Discussion

The prepared composites were examined in both solution and solid phase to inspect the influence of the ceramic nanofillers on the microstructure and thermal, optical, and dielectric properties of the chitosan-derived composites.

### 3.1. Characterisation of the BT Powders

The phase composition of ultrafine BT powders was investigated by X-ray diffraction at room temperature. It can be observed in Figure 1 that the powders were mainly the perovskite phase, with minor traces of BaCO_3_, due to the alkaline medium of the reaction solution, even though the vessel was closed all the time. As can be seen from the peak at 2θ = 45°, the powder had a pseudo-cubic symmetry Pm-3m, which was indexed with PDF card.0-0213. The TEM image (inset Figure 1) shows that the BT powder was homogeneous in size and shape with a spherical shape and an average diameter of 20 nm. Furthermore, the powders were not very agglomerated, which allowed us to achieve a good dispersion in chitosan and a good homogeneity of the composite, as will be revealed in the next part of the paper.

These results are in good agreement with previous papers [46,47], where it was shown that the precipitation method allows the obtaining of spherical particles with controlled size, and if the particles are smaller than 100 nm, BT will have cubic symmetry [48,49,50].

### 3.2. Shear Oscillatory Testing of the Matrix and Composite Solutions

The ceramic particles were introduced in the chitosan matrix solution, and the corresponding chitosan/BT dispersions were subjected to rheological analysis. Experiments were undertaken in oscillatory mode because the viscoelastic properties of the samples are known to be affected by the dispersion level of the nanoparticles and interactions from the system (i.e., matrix–filler and filler–filler) [39]. For all samples, the frequency sweep data were registered at constant shear strain in the linear viscoelastic regime (LVR). Figure 2a–f depicts the variation of the elastic (G′) and viscous (G″) shear moduli with shear frequency at 25 °C for the samples containing variable nanofiller amounts. The gradual incorporation of nanoparticles produced the increase in the G′ and G″. The presence of the rigid inorganic filler enhanced the elasticity of the composite (higher G′), while the viscous component was increased, owing to greater dissipation in the sample because of the friction among the nanofiller particles [51].

The oscillatory measurements reveal that all samples displayed frequency-dependent behavior of G′ and G″, and two different zones were noticed at extreme frequencies. At low shearing, one may notice that G″ > G′ and, as the shearing frequency increased, the rheological module grew until they became equal (G″ = G′), and finally, the variation was reversed (G′ > G″). Hence, the samples underwent a transition from liquid-like (G″ > G′) to solid-like (G′ > G″) behavior. As shown in Figure 2, the crossover frequency where G″ = G′ (fc) became higher as the BT amount in chitosan increased from 0.53 Hz (matrix solution) to 1.69 Hz for the dispersion containing 10% vol. BT.

On the other hand, at high shearing, the rheological response of the composite did not differ so much since the shear moduli tended to converge. This emphasizes the fact that the flow properties of samples at high frequencies were dominated by the continuous phase (chitosan matrix), while the influence of the nanofillers was less relevant. Similar properties were reported for other types of polymer composites [52,53]. When analyzing the oscillatory data at small frequencies, one may see a higher slope of the elastic modulus plot, whereas this aspect was less pronounced for the viscous modulus curve. The invariability of the rheological moduli to shear frequency denotes the prevalence of the particle–particle interactions at lower frequencies [52]. Frequency-independent characteristics were also demonstrated for other loaded polymer systems [53,54]. For loaded polymers, the literature confirms that the elastic modulus reflects better structural changes than the viscous modulus [55]. The comparison of the frequency behavior of chitosan/BT systems at several loading levels is presented in Figure 3a, where the G′ against shear frequency is plotted. It is observed that under 1 Hz, the plots of the elastic modulus depended on the frequency for the matrix solution and dispersion containing 0.5% vol., while starting with 1% vol. BT in the system G′ curves were independent on the frequency, denoting a powerful solid-like behavior. Additional information can be extracted by examining the variation of the storage modulus taken at 0.1 Hz with sample composition (see Figure 3b).

The classical percolation theory was employed to clarify how gradual incorporation of nanofiller contributes to changes in the viscoelastic behavior of the prepared samples due to the occurrence of a percolated network at a critical composition. Hence, the rheological PT was evaluated by using Equation (1) [41]:G′ = G_0_ (φ_f_ − φ_c_)^γ^,(1)
where G_0_ is a proportionality constant; φ_f_ is the filler volume fraction; φ_c_ is the critical volume fraction at the PT; and γ is the critical exponent, which is linked to network dimensional features.

Equation (1) works for compositions above the percolation threshold, when φ > φ_c_. To assess the rheological PT based on the oscillatory data at variable nanofiller contents, the experimental plot was examined based on the double-logarithmic relation: log G′ against log(φ − φ_c_), where the value of the φ_c_ is changed until the best linear fit is attained (see inset from Figure 3b). The resulting φ_c_ value was of 0.9% vol. of BT nanoparticles in chitosan, while the value of the critical exponent was γ = 1.0938. This value agrees with the literature [56].

### 3.3. Morphological Analysis of the Films

To elucidate the dispersion of BT particles within the chitosan matrix, the BT/chitosan nanocomposite films were examined in fracture using scanning electron microscopy. The SEM micrographs at both 1500× and 25,000× magnification are displayed in Figure 4. Efforts to enhance the magnification above this threshold yielded pictures that were blurry and ambiguous. Consequently, for better visualization, a 50% sharpen correction was applied to the 25,000× SEM images. The morphology of the chitosan matrix (Figure 4a) was compact, uniform, smooth, and homogeneous, with few defects and no visible inclusions, thus lacking a granular texture. These characteristics are specific to pure chitosan, which does not contain dispersed phases or inorganic inclusions. Following that, it can be observed that the BT/chitosan nanocomposite films were dense, without air pores, regardless of composition (Figure 4b–d). Moreover, due to the very small size of the BT particles (indicated by red arrows), the dispersion within the films was homogeneous, without a compositional gradient or agglomeration, as observed in other studies [57]. For the BT/chitosan nanocomposite containing 2.5% vol. BT (Figure 4b), fine unevennesses emerged, suggesting an incipient dispersion of barium titanate particles. Their distribution seemed relatively discontinuous considering the low concentration of filler. On the other hand, the surface of the composite film including 5% vol. BT (Figure 4c) exhibited morphological balance, characterized by a fine granular texture without obvious agglomerations, indicating an adequate particle dispersion within the biopolymer matrix and an efficient interaction between the chitosan phase and the inorganic phase. Figure 4d highlights a visibly rougher and slightly more irregular surface, where increasing agglomeration of distinct particles can be observed due to the higher concentration of dispersed inorganic material (10% vol. BT). This uniform dispersion of particles creates many polymer/particle interfaces that can have a crucial role in the functional properties of the composites.

### 3.4. FTIR Characterisation

For a better understanding of the BT–chitosan interaction, the FTIR analysis was employed. As shown in Figure 5a, the typical vibrations of neat chitosan were those of primary amine and the polysaccharide backbone, which were superposed by the residual acetylated part. We distinguished the ν(NH_2_) and δ(NH_2_) vibrations at 3361/3286 cm^−1^ and 1640 cm^−1^, respectively; the strong ν(COC) + ν(C-OH) at 1065/1027 cm^−1^; and the contribution of the amide II band to the 1551 cm^−1^ signal [58,59]. The FTIR spectrum of the composite containing 10% vol. BT was dominated by two intense absorption bands that covered overlapped signals from protonated primary amine from chitosan, undisturbed primary amine and amide I band, and acetate ions. Two types of carboxylate groups can be potentially generated in chitosan−BaTiO_3_ + acidic acid mixture: (i) deprotonated acetic acid in interaction with the primary amine from chitosan, leading to protonated NH_3_^+^ units, and (ii) barium acetate, which can be produced in the chemical reaction between BaCO_3_ (the BaTiO_3_’s impurity) and acetic acid (as a solvent). By a curve-fitting procedure with a Voigt profile, the individual contribution of each species to the whole absorption is presented in Figure 5b.

Therefore, the antisymmetric and symmetric stretching vibrations of NH_3_^+^ units were observed at 1585 and 1534 cm^−1^, respectively, and the antisymmetric and symmetric stretching vibrations of carboxylate COO^−^ units were detected at 1549 and 1407 cm^−1^ [58,59,60]. The FTIR findings can be described by the following reaction mechanism (Figure 6):

### 3.5. Thermal Behavior

To understand how the temperature affects the properties of the chitosan and chitosan/BT films, TGA and DSC experiments were performed. The TGA and DTG curves for the samples are displayed in Figure 7a,b, respectively.

One may see that for the neat chitosan, the weight loss was developed in two stages. The first one began at around 66 °C and led to weight loss of 10% at ~112 °C and can be attributed to the loss of moisture. The matrix structure was composed of polar hydroxyl and amine groups, which enabled the interaction with water molecules. So, chitosan-based materials inevitably contain an arbitrary and small amount of water that can act as plasticizer [61,62]. The second stage was marked at 240 °C and evolved towards a maximum of 345 °C with a weight loss of 42.3%. In this temperature interval, weight loss is caused by glucosamine chains degradation under heating, vaporization, and elimination of volatile products [63]. Similar data are reported in other studies concerning thermal behavior of chitosan [62,64]. Nieto et al. [65] reported that pyrolysis of such natural polymers begins with arbitrary splitting of the glycosidic linkages, which is continued by deeper decomposition into acetic and butyric acids and other compounds where C2, C3, and C6 are prevalent [64]. The thermal degradation of the prepared composites is found to be shifted to a larger temperature and left more residue. The insertion of nanoparticles in the chitosan enhanced thermal stability due to the polymer–nanofiller interactions and the heat resistance of the ceramic phase. This was also observed for other polymer/BT composites [66].

DTG curves present similar shapes for all samples. Polysaccharides are widely known to have affinity for water and upon hydration they acquire disordered structures. Close examination of Figure 7b indicates that there were small variations in peak area and position, showing that the samples differed in moisture absorption and interactions in the system due to variable BT loading in the chitosan. Regardless of the content of the inorganic phase, the specimens had a main degradation peak in the temperature interval of 163–370 °C. The values of the maximum decomposition temperature rate (T_max_) data for the studied materials are summarized in Table 1. The pure polymer exhibited a thermal event at the maximum temperature of 276 °C. The addition of the ceramic nanoparticles in the chitosan determined a slight increase in the T_max_ parameter, between temperature values corresponding to that of the matrix and composite with 2.5% filler. Such an increase in the degradation temperature was caused by enhanced physical interactions among the composite phases, which hindered the mobility of chitosan chains. An additional contribution to this aspect resided in the retardant effect of the BT nanofillers [67]. Moreover, the good dispersion of the ceramic nanoparticles may limit and reduce the evolvement of the degradation products from the matrix towards the gaseous phase, postponing the process of thermal degradation. This means that the incorporation of BT slightly increases the decomposition temperature of the samples, conferring the composites good thermal stability.

The DSC thermograms, presented in Figure 8a, provide relevant information on the transition temperatures changes in the samples. The maximum temperature of 180 °C was chosen to avoid potential polymer degradation. Also, to avoid moisture effects on the glass transition temperature (T_g_), the data from the second heating run were employed. According to literature data [68,69], polysaccharides show no melting tendency, but they degrade because of heating; hence, these biopolymers are thermally degrading before melting. Consequently, the DSC plots from Figure 8a showed no endothermic melting profile. As seen in Figure 8b, the loaded polymer samples displayed no noteworthy variations of the T_g_ values with composition. It is noted that the gradual introduction of BT nanoparticles in chitosan produced a small increase in the T_g_ parameter. This is analogous to the literature [70], which demonstrates that BT determines the reduction of the macromolecular chain mobility in the composites, owing to the diminishment of the free volume.

### 3.6. Optical Properties

An insight into the effect of the BT nanofillers on the optical properties of the chitosan was given by UV-VIS spectroscopy data. As observed in Figure 9a, the transmittance of the samples gradually decreased in the visible domain due to the addition of the ceramic particles. The spectrum of the polysaccharide matrix exhibited a sharp absorption edge, which was marginally changed upon insertion of the BT particles. This is supported by reports dealing with optical characteristics of BT-containing composites [71,72]. Also, the presence of the inorganic nanoparticles in chitosan created many interfaces that were responsible for light scattering and augmented the absorption. The interaction of the electromagnetic radiation with the prepared samples can be estimated via absorption coefficient (α), which is defined by Equation (2):α = (1/t) ln(1/T),(2)
where t denotes the thickness of the film and T denotes the transmittance.

As remarked upon in Figure 9b, the magnitude of α was augmented as the content of ceramic phase was higher in the composites, comparatively with neat chitosan. The spectra of the absorption coefficient can be analyzed with the help of Davis–Mott theory [73] to understand optical transitions and to estimate some optical parameters. The plots of α versus photon energy (E) from Figure 9b display different domains, where the slope changed with variation of the sample composition. By linearization of the Urbach rule [74] expressed by Equation (3), one may attain information on the Urbach energy (E_U_):α = C exp(E/E_U_),(3)
where C is a constant.

The values of E_U_ are determined from the reverse of the slope of the absorption coefficient dependence on the photon energy. The absorption at energies under the optical gap describes the Urbach edge. The inset from Figure 9b shows that the magnitude of E_U_ was increased after the incorporation of the BT nanofillers in the matrix. Thus, the presence of the inorganic phase contributed to the broadening of the intrinsic absorption edge of samples, owing to the created disorder within the composites, and local electric fields became more intense. This aspect can be linked to transitions from the extended valence band states to the localized states at the conduction band tail.

In the high absorption domain, the spectra of α can be related to interband transitions. In this region, it is possible to assess the optical band-gap energy (E_g_) based on the Davis–Mott expression [73], as shown in Equation (4):αE = C_0_ (E − E_g_)^x^,(4)
where C_0_ is a constant and x is an index depicting the type of electronic transition.

To evaluate the optical band-gap energy corresponding to the direct (E_g,d_) and indirect (E_g,i_) transitions, the graphical representation of (αE)^1/x^ against the incident photon energy was performed by selecting x = 0.5 and x = 2. Linear zones can be distinguished in all plots, and their intersection with x-axis serves for the evaluation of both E_g,d_ and E_g,i_, as indicated in Figure 10a,b. The changes of E_g,d_ and E_g,i_ with the sample composition are illustrated in the insets from Figure 10 (images (a’) and (b’)). The introduction of the BT nanoparticles is responsible for creating localized states between the valence and conduction bands and this reduces the band gap energy corresponding to either direct or indirect transitions. The data achieved show that E_g,d_ was higher than E_g,i_ at all loading levels in the chitosan. Similar effects on E_g_ due to BT addition in polymers were previously reported for other composite materials [71,72].

Another step in optical analysis relies on the refractometric evaluation of the samples. The speed of optical radiation through the chitosan-based specimens can be expressed via the refractive index (n). This optical parameter was measured at 589 nm and at a constant temperature of 25 °C, with the recorded data being listed in Table 2. Neat chitosan film presented a refractive index of 1.5163, which agreed with that reported in the literature [75]. One may observe that the incorporation of the BT nanoparticles in the biopolymer was responsible for the enhancement of n, owing to their high polarity. At high filler concentrations, the material’s density and the size of polar domains became larger, so that the refractivity was found to be in continuous increase. This is in accordance with the literature [76].

Measuring the refraction can be regarded as a non-electrical method of evaluating the electrical properties at optical frequencies. Based on the absorption features, one may estimate the extinction coefficient (k_ext_), which quantifies the material potential to absorb or reflect radiation of definite energy. According to Table 2, the magnitude of k_ext_ was affected by the loading level of chitosan, revealing that light was absorbed to a higher extent when more BT nanoparticles were inserted into the matrix. Furthermore, the refractive index and extinction coefficient were paramount for determining key electrical properties in optical frequencies regime, namely, optical conductivity (σ_opt_) and real (ε_opt_′) and imaginary (ε_opt_″) parts of the optical dielectric constant, as indicated by the relations (5)–(7):σ_opt_ = (α n c_e_)/4π,(5)ε_opt_′ = n ^2^ − k_ext_^2^(6)ε_opt_″ = 2 n k_ext_,(7)
where c_e_ is the speed of light in empty space.

Optical conductivity exposes the material’s optical reaction brought on by the charge carriers’ movement under the effect of light. Incorporation of the inorganic nanoparticles appeared to significantly favor greater carrier transport within the biopolymer, given that σ_opt_ increased by one order of magnitude. Regarding the dielectric characteristics at optical frequencies, one may notice that ε_opt_′ was governed by the refractivity, whereas ε_opt_″ was influenced by the extinction properties of the samples. Data from Table 1 indicate that the ε_opt_″ progressively increased from 3.41 × 10^−4^ (neat chitosan) to 0.0104 (composite with 10% vol BT), but the dielectric losses were low. Additionally, the real part of the dielectric constant was higher when the content of the ceramic phase in the composite was bigger, so ε_opt_′ increased by about 1.07 times compared to the matrix. The limitation in this variation with the sample’s composition was linked to the fact that at such high frequency, only electronic polarization contributed to the overall dielectric behavior.

### 3.7. Impedance Spectroscopy Analysis

The impedance spectroscopy analysis results performed at room temperature on the chitosan and chitosan/BT films are shown in Figure 11a. In the case of pure chitosan film, only one semicircle was visible (inset of Figure 11a), whereas for all composites, two distinct semicircles can be identified: (i) one at high frequencies linked to the intrinsic dielectric properties of the material, which shifted towards lower frequencies with the addition of BT; (ii) the second was an incomplete semicircle characteristic of low frequencies, mainly associated with interfacial polarization and electrode effects. The second semicircle was incomplete because of the limited frequency range. A complete semicircle necessitated a lower frequency. The equivalent circuit proposed for describing impedance data for composite films is presented in Figure 11b.

Fitting the impedance response to the proposed equivalent circuit allows for the extraction of information regarding the resistive and capacitive responses of the components, which can subsequently be analyzed in relation to the electrical properties of the materials.

The experimental complex impedance responses, represented as Argand diagrams, along with the fitting semicircles, are illustrated for the extreme compositions (0.25 and 10% vol.) in Figure 12a,b and for all other compositions in Appendix A. The basic model correlated each semicircle in the complex impedance spectrum with a series-arranged RC parallel circuit response. The resistance value was determined by the intersection of each semicircle with the Re(Z) axis, while the capacitance value can be calculated from the frequency corresponding to the maximum of the semicircle and was obtained from impedance spectroscopy measurements [77].

The low frequency component, which was not fully represented in the measured frequency range, had f_max_ set at 1.5 Hz. Upon careful examination of the axis origin, none of the complex impedance graphs started exactly at the coordinates (0, 0). This observation reveals that all analyzed samples displayed an inherent non-zero resistance in the electric circuit. This resistance, noted as R_s_, is associated with their direct current conductivity [77]. The results of the fits of the complex impedance response are summarized in Table 3.

The description by equivalent circuits of impedance response of composite systems is not unique, and to assign these circuit values to the material properties (capacitance and conductivity) is not a simple task. The observed general trend of the equivalent circuit elements when increasing the filler amount shows a few clear features:A tendency to reduce the series resistance Rs (i.e., to reduce the overall composite dc-resistivity) when increasing the BT filler amount, due to the good dielectric properties of BT powders, which strongly limit the dc-conduction paths in chitosan; these results agree with previously observed ones [78,79,80].The high-frequency components R1 and C1, assigned to the intrinsic material’s properties (resistivity and conductivity), present a strong composition-induced effect: (i) up to 2.5% vol. BT the resistance R_1_ remained almost constant at 0.2 MΩ, while its capacitance increased almost two times; (ii) for higher BT concentration (>2.5% vol. BT), the resistance R_1_ strongly increased together with a reduction in capacitance.At low-frequency response (R_2_, C_2_), circuit elements were affected both by the presence of BT filler and electrode effect: thus, for BT addition ≤ 2.5%, a decrease in R_2_ and an increase in C_2_ was observed, while for larger BT amount, an increase in R_2_ and a decrease in C_2_ was obtained.

Since these circuit elements cannot discriminate between the contribution of the electrodes, interfaces, or composition, the only conclusion is that at 2.5% vol. BT, the electric response of the composite films changed dramatically. This change in electrical behavior could also be an indication of BT particle percolation, and only a detailed dielectric analysis can confirm or deny this.

### 3.8. Frequency Dependence of Dielectric Properties

In Figure 13a,b, the frequency dependencies of the real and imaginary parts of dielectric permittivity for chitosan film and chitosan/BT composites at room temperature are shown. Due to chitosan’s hygroscopicity, the films were dried in an oven at 70 °C/2 h before the measurements and kept in a desiccator during electrical experiments. At low frequencies, all the samples showed both a Debye-type dipolar relaxation and a Maxwell–Wagner-type relaxation. This Maxwell–Wagner-type relaxation is attributed to electrode effects and interfacial polarization [78,79,80] and now is more visible due to large number of interfaces between small BT particles and the polymer. The intrinsic dielectric properties of the material were revealed only for frequencies above 1 kHz. When BT was added, the effective permittivity increased (Figure 13c) from 6 for the pure chitosan film to about 11.5, while the dielectric losses decreased. For low BT content (0.25–1%), the increase was nearly linear, but for higher content (>2.5%), saturation tendency can be observed in agreement with previous theoretical calculations [81] and with impedance spectroscopy data (Section 3.7). Although permittivity had this saturation tendency, the dielectric losses continued to decrease, which led to an overall improvement in the dielectric behavior of chitosan/BT composites.

The dielectric losses of the composites (Figure 13d) exhibited a maximum as a function of frequency for all samples, which is associated with the α- relaxation of chitosan [79,80]. In this case, the maximum shifted with the increase in BT amount from 200 Hz to lower frequencies (~50 Hz), indicating an increase in the system relaxation time (from 0.6–0.7 ms for composites with low BT content (≤1%) up to 3 ms for composites with 10% BT). These results are in contrast with our previous study [78], where the relaxation time decreased with BT addition and meant that ultrafine BT particles delayed the relaxation mechanism in the polymer matrix due to the large number of interfaces. Also, this trend of increasing relaxation time can be correlated with PT of BT particles from the rheological measurements (Section 3.2) where a concentration of 0.9% vol. causes a change in the behavior of composite solutions.

### 3.9. High Field Properties

After the complete characterization at low field, the sample with 5% vol. BT (~38% mass BT) was selected for investigation at high field. Due to significant losses (Figure 13d), a new experiment was designed to perform high-voltage measurements: the dc-high voltage was applied in 10 min sequences, after which the frequency dependencies of the dielectric properties were measured. This way, the effect of the electric field on the dipolar and Maxwell–Wagner relaxation phenomena that was observed in low field (Section 3.8) was investigated. The results are presented in Figure 14.

From Figure 14a,b, it can be observed that the high field determined a decrease in both the real and imaginary parts of permittivity and a shift of dipolar relaxation to higher frequency. At the same time, the values for the applied electric field at which a variation began to be observed were very high (~400 kV/cm) compared to the values reported in our previous study [77], but in accordance with those reported in the literature for chitosan-based nanocomposites [44,82]. These very high values are required to observe nonlinear phenomena due to the combined effect of the polymer with the very small sizes of the BT particles [81]. However, for applied fields larger than 500 kV/cm, a strong variation (almost linear) of permittivity with applied fields was obtained, together with a reduction of dielectric losses (Figure 14c,d). In the inset of Figure 14c, the calculated tunability (n = ε(0)/ε(E)) at the selected frequencies is represented. This time, unlike in our previous papers, this was an indirect method to determine tunability because the variation of permittivity with the applied electric field was not measured directly [77,82] but determined from impedance dielectric spectroscopy. This is one of the reasons for smaller tunability values [77]. Another reason is the small particle size of BT and its cubic symmetry that dramatically reduced dielectric nonlinearity [83]. However, extrinsic effects (multiple polymer–ceramic particle interfaces) allowed for the obtaining of a considerable variation of the permittivity with the applied field (n~2.4 at E = 660 kV/cm and f = 550 kHz), together with a diminution of the α-relaxation phenomena of the polymer (Figure 14d) that makes these composites suitable materials for microelectronics applications.

## 4. Conclusions

Ultrafine BT particles (~20 nm) were synthesized by a coprecipitation method and then used for preparation of chitosan/BT films with different BT fractions (0–10% vol.). A uniform dispersion of the BT particles was observed in all composite films. The PT of BT particles in chitosan-based composites was determined both by rheological measurements and dielectric spectroscopy. The results agreed with the literature, with a lower PT from rheology (~0.9% vol. BT) compared to that from electrical measurements of ~1–2.5% vol. BT. The thermal analysis data revealed that the insertion of the nanoparticles is beneficial for improving the thermal resistance of the chitosan nanocomposites since the BT filler generated a slight increase in the Tg and of the temperatures corresponding to 5 and 10% weight losses.

The electric response of the composite materials was examined using the impedance spectroscopy technique in relation to analogous electric circuits. The dielectric characteristics as a function of frequency indicated an increase in permittivity (about 45% for 10% BT filler) accompanied by a significant reduction in dielectric losses with BT additions. The nonlinear properties for 5% vol. BT showed a considerable variation of the permittivity at a higher applied field (n~2.4 at E = 660 kV/cm) together with a diminution of the α- relaxation phenomena of the polymer. This is a very promising result in using these composites as tunable flexible structures due to the possibility of controlling both permittivity and dielectric losses with electric fields. The results concerning the strong reduction of polymer relaxation under electric field were produced for the first time.

## Figures and Tables

**Figure 1 polymers-17-01612-f001:**
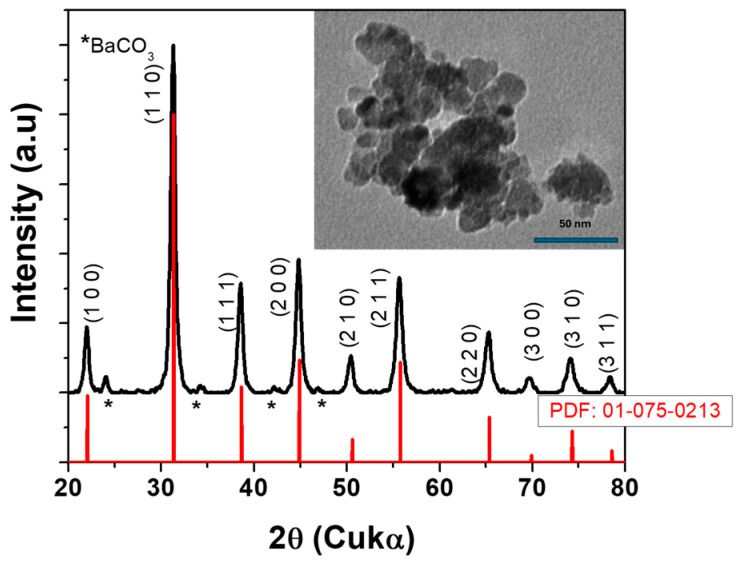
XRD pattern for BT particles (inset—TEM image of particles); * BaCO_3_ the secondary phase.

**Figure 2 polymers-17-01612-f002:**
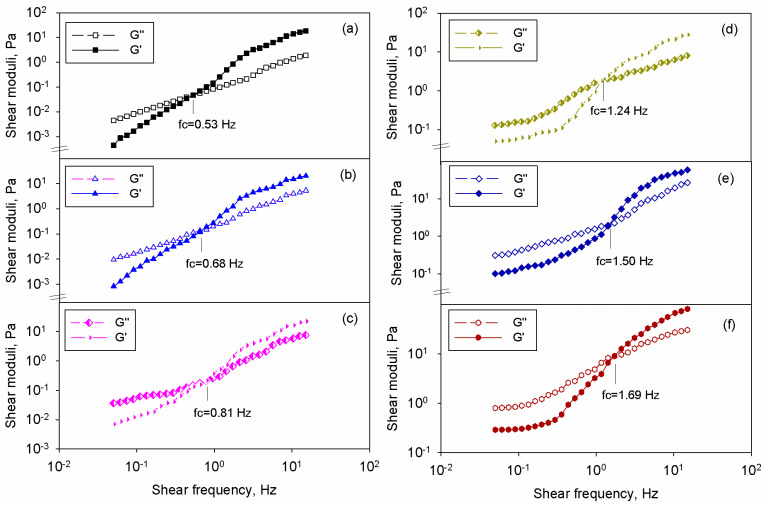
Frequency sweep tests for (**a**) chitosan solution and the corresponding dispersions containing (**b**) 0.5% vol. BT, (**c**) 1% vol. BT, (**d**) 2.5% vol. BT, (**e**) 5% vol. BT, and (**f**) 10% vol. BT.

**Figure 3 polymers-17-01612-f003:**
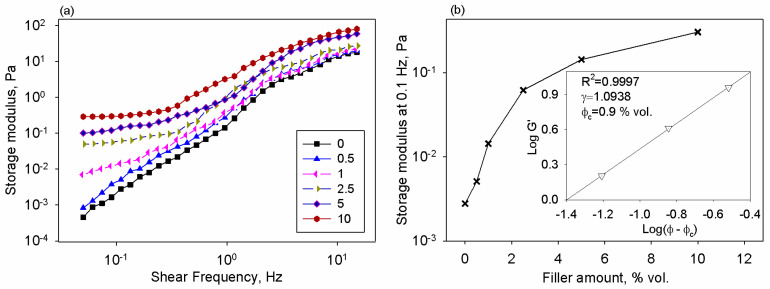
(**a**) Variation of the shear moduli with frequency for all samples, and (**b**) dependence of elastic modulus at 0.1 Hz on the filler content in the chitosan-based systems. The inset graph depicts log G′ against log(φ − φ_c_).

**Figure 4 polymers-17-01612-f004:**
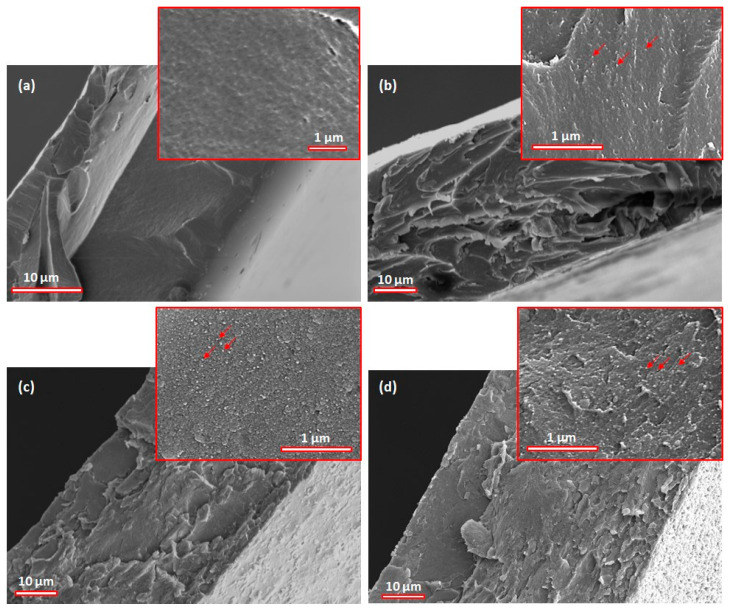
SEM images on fracture of chitosan matrix (**a**) and BT/chitosan nanocomposite films: 2.5% vol. BT (**b**), 5% vol. BT (**c**), and 10% vol. BT (**d**). Red arrows indicate the BT particles.

**Figure 5 polymers-17-01612-f005:**
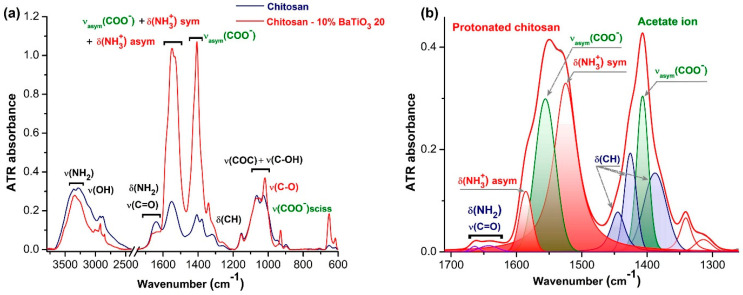
Infrared analysis of chitosan—BT-20 films: (**a**) ATR-FTIR infrared spectra of doped chitosan as compared to native chitosan; (**b**) spectral decomposition of the complex bands in the 1700–1250 cm^−1^ domain.

**Figure 6 polymers-17-01612-f006:**
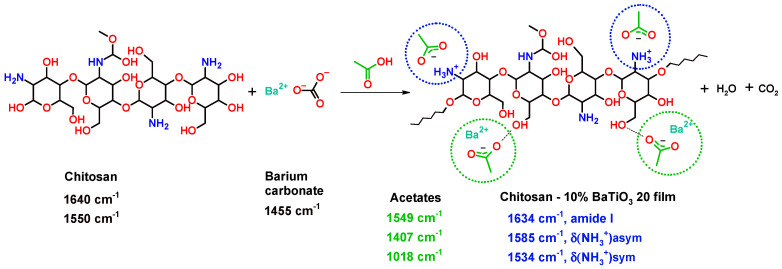
Reaction mechanism proposed by FTIR analysis.

**Figure 7 polymers-17-01612-f007:**
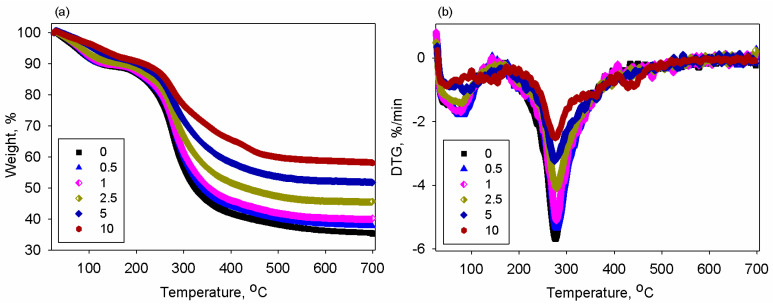
(**a**) TGA and (**b**) DTG curves of the chitosan samples containing variable BT content.

**Figure 8 polymers-17-01612-f008:**
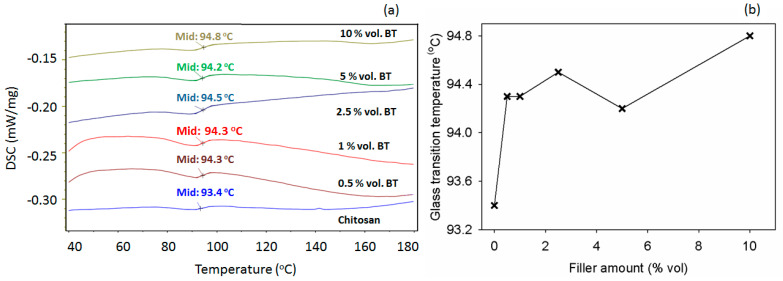
(**a**) DSC thermograms of the chitosan and chitosan/BT composites and (**b**) variation of glass transition temperature with the BT content in the samples.

**Figure 9 polymers-17-01612-f009:**
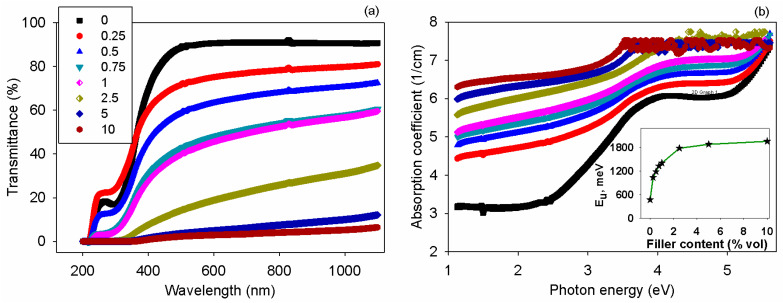
(**a**) UV-VIS spectra and (**b**) absorption coefficient of the chitosan samples with variable BT content. Inset graph represents Urbach energy against filler content in the samples.

**Figure 10 polymers-17-01612-f010:**
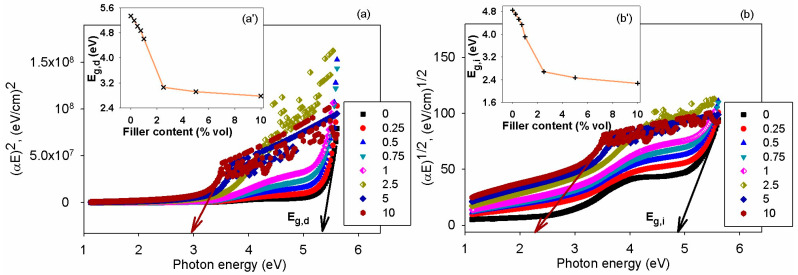
The dependence of (**a**) (αE)^2^ and (**b**) (αE)^1/2^ on E for the chitosan samples with variable BT content. The insets denote the variation of (**a’**) E_g,d_ and (**b’**) E_g,i_ with filler amount in the system. The arrows indicate the **x**-**intercept** of linear part of the Tauc plot where the direct/indicrect band gap is obtained.

**Figure 11 polymers-17-01612-f011:**
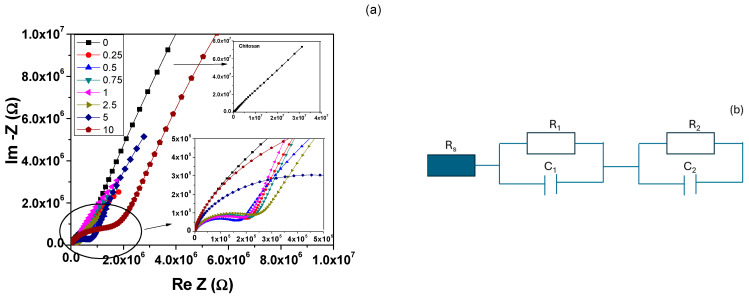
(**a**) Experimental Argand diagrams: Insets (up)—pure chitosan films; (down)—zoom on high frequency region. (**b**) Equivalent circuits proposed to describe the complex impedance response of composite films.

**Figure 12 polymers-17-01612-f012:**
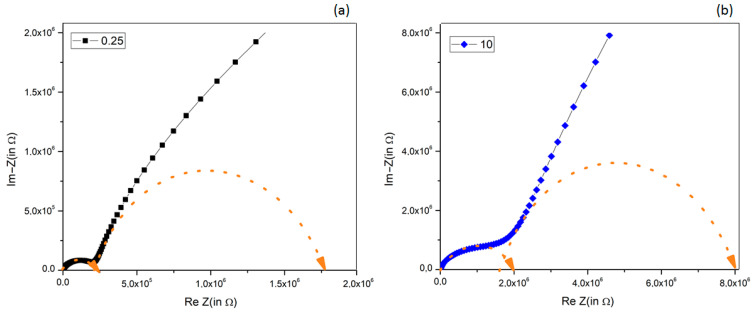
Fitting experimental Argand diagrams with equivalent circuits for (**a**) 0.25% BT and (**b**) 10% BT. The dashed arrows represent the semicircles used for fitting experimental data.

**Figure 13 polymers-17-01612-f013:**
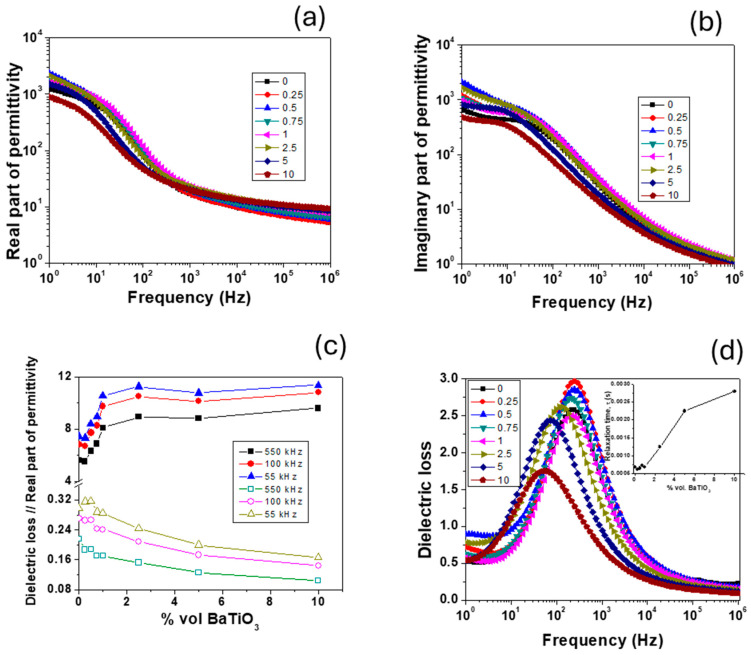
Dielectric properties of chitosan/BT films: (**a**,**b**) frequency dependence of real and imaginary part of permittivity; (**c**) composition dependence of dielectric losses and real part of permittivity at few selected frequencies, and (**d**) frequency dependence of dielectric losses (inset—composition dependence of relaxation time).

**Figure 14 polymers-17-01612-f014:**
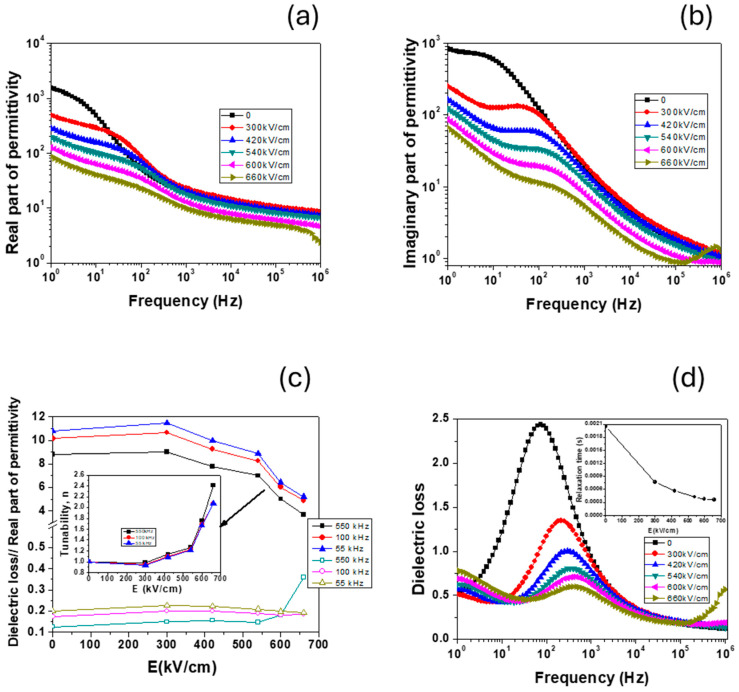
High field dielectric properties of 5% vol. BT: (**a**,**b**) frequency dependence of real and imaginary part of permittivity at the few selected applied dc-fields; (**c**) field dependence dielectric losses and real part permittivity at the few selected frequencies (same frequency as in Figure 13c)—Inset—dc-tunability vs. applied electric field; (**d**) frequency dependence of dielectric losses (inset—field dependence of relaxation time) at the few selected applied dc-fields.

**Table 1 polymers-17-01612-t001:** Thermogravimetric data of the chitosan and chitosan/BT samples: temperatures corresponding to 10% (T_10_) and 50% (T_50_) weight losses, maximum decomposition temperature rate (T_max_), and the residual char at 700 °C (R).

Filler Content, % vol.	T_10_, °C	T_50_, °C	T_max_, °C	R,%
0	112	321	275.64	35.15
0.5	127	340	278.41	37.76
1	141	358	277.82	40.30
2.5	178	426	278.50	45.67
5	202	-	274.28	51.81
10	211	-	276.32	58.07

**Table 2 polymers-17-01612-t002:** Data extracted at 589 nm for the refractive index, absorption coefficient, extinction coefficient, optical conductivity, and real and imaginary parts of the optical dielectric constant of the samples.

Filler Content, % vol.	n	α (cm^−1^)	k_ext_	σ_opt_ (s^−1^)	ε_opt_′	ε_opt_″
0	1.5163	23.9963	1.1253 × 10^−4^	0.87 × 10^11^	2.2992	3.4126 × 10^−4^
0.25	1.5221	115.9805	5.4389 × 10^−4^	4.21× 10^11^	2.3168	1.6557 × 10^−3^
0.5	1.5262	171.6537	8.0497 × 10^−4^	6.25 × 10^11^	2.3293	2.4571 × 10^−3^
0.75	1.5277	219.0769	1.0274 × 10^−3^	7.98 × 10^11^	2.3339	3.1390 × 10^−3^
1	1.5286	257.3681	1.2069 × 10^−3^	9.39× 10^11^	2.3366	3.6898 × 10^−3^
2.5	1.5347	423.2049	1.9846 × 10^−3^	1.55 × 10^12^	2.3553	6.0916 × 10^−3^
5	1.5533	573.3283	2.6886 × 10^−3^	2.13 × 10^12^	2.4127	8.3525 × 10^−3^
10	1.5704	706.7173	3.3141 × 10^−3^	2.65 × 10^12^	2.4661	0.0104

**Table 3 polymers-17-01612-t003:** Fitting parameters obtained from complex impedance spectra for composites films.

Sample	R_s_ (Ω)	R_1_ (MΩ)	C_1_ (nF)	f_max1_ (Hz)	R_2_ (MΩ)	C_2_ (nF)	f_max2_ (Hz)
0.25	230	0.25	0.35	1850	1.67	63.5	1.5
0.5	191	0.22	0.44	1585	1.41	75.2	1.5
0.75	176	0.22	0.52	1360	1.34	79	1.5
1	150	0.21	0.64	11,666	1.34	79	1.5
2.5	124	0.28	0.76	735	1.12	94	1.5
5	109	0.82	0.35	541	2.13	49	1.5
10	89	1.98	0.32	251	5.7	18	1.5

## Data Availability

The original contributions presented in this study are included in the article/Appendix A. Further inquiries can be directed to the corresponding authors.

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
