# Peer review of "Design of Polysaccharide-Based Nanocomposites for Eco-Friendly Flexible Electronics"

_polymers, 2025, doi:10.3390/polym17121612_

Round 1
Reviewer 1 Report
Comments and Suggestions for Authors
This work reported the obtaining of chitosan/ceramic BT and a technique for producing a large quantity of ultrafine barium titanate powders at ambient temperature. The results demonstrated a straightforward method for producing tunable flexible structures. The manuscript is well-structured with sound logic, sufficient evidence, and notable innovation; therefore, I recommend its acceptance after minor revisions.
- The introduction section would benefit from more concise wording and better logical flow through improved paragraph structuring to enhance readability.
- The XRD analysis should include the crystal phase identification of BaTiO3 along with the corresponding standard reference card (e.g., JCPDS/ICDD PDF card) for proper phase verification and comparison.
- The EIS analysis should be supplemented with the corresponding equivalent circuit diagram to facilitate proper data interpretation and modeling of the electrochemical interface processes.
- The manuscript requires additional formatting corrections, including: (1) adding frames/borders to Figures 6a and 10 to maintain consistency with other figures, and (2) standardizing the position of all figure numbers throughout the document for uniform presentation.
Author Response
Dear reviewer,
The authors would like to thankful for the careful analysis of the manuscript and the valuable observations and the requested changes were marked with red in the manuscript.
- The introduction section would benefit from more concise wording and better logical flow through improved paragraph structuring to enhance readability.
Response 1: The introduction was revised to render enhanced readability, as indicated by the reviewer.
- The XRD analysis should include the crystal phase identification of BaTiO3 along with the corresponding standard reference card (e.g., JCPDS/ICDD PDF card) for proper phase verification and comparison.
Response 2: Figure 1 was completed with PDF card used for identification of BaTiO3 crystalline phases.
- The EIS analysis should be supplemented with the corresponding equivalent circuit diagram to facilitate proper data interpretation and modeling of the electrochemical interface processes.
Response 3: Figure 11 with EIS was completed with the equivalent circuit diagram.
- The manuscript requires additional formatting corrections, including: (1) adding frames/borders to Figures 6a and 10 to maintain consistency with other figures, and (2) standardizing the position of all figure numbers throughout the document for uniform presentation.
Response 4: All figure numbers were standardized for uniform presentation.
Reviewer 2 Report
Comments and Suggestions for Authors
The manuscript titled “Design of Polysaccharide-Based Nanocomposites for Eco-Friendly Flexible Electronics” by Turcanu et.al is a well-organized study on the development of chitosan-based nanocomposites reinforced with barium titanate nanoparticles for potential eco-friendly flexible electronics. The research is original and provides important insights into sustainable materials for electronic applications. The authors have successfully synthesized ultrafine BT nanoparticles and integrated them into a chitosan matrix, demonstrating tunable properties through comprehensive characterization. The study addresses key challenges in the field, such as the demand for cost-effective, non-toxic, and flexible electronic components. I recommend the acceptance of the manuscript for publication in Polymers journal after the authors address the following points:
- The authors can clarify the exact parameters for BT nanoparticle synthesis (e.g., pH control during precipitation, drying conditions).
- The SEM images in Figure 4b at 2.5% BT, they mention "incipient dispersion," but the resolution doesn't clearly show nanoparticle distribution. Higher magnification in SEM or TEM measurement for the composite is beneficial.
- Authors should confirm the interfacial bonding between BT and chitosan by using IR or XPS (better option).
- The impact of water absorption on dielectric properties is underexplored. Chitosan’s hygroscopicity may affect long-term stability and should be addressed.
- In the Conclusion part, the authors should emphasize the relation between dielectric performance and sustainability to highlight the novelty of the work.
Author Response
The authors would like to thankful for the careful analysis of the manuscript and the valuable observations and the requested changes were marked with red in the manuscript.
- The authors can clarify the exact parameters for BT nanoparticle synthesis (e.g., pH control during precipitation, drying conditions).
Response 1: We added in Paragraph 2.3 details about the synthesis conditions:
“The concentration of the NaOH solution allows maintaining the pH=14 during precipitation. After, the suspension was washed with distilled water until the pH becomes neutral. Finally, the obtained wet powder was frozen and lyophilized for 24 hours.”
2. The SEM images in Figure 4b at 2.5% BT, they mention "incipient dispersion," but the resolution doesn't clearly show nanoparticle distribution. Higher magnification in SEM or TEM measurement for the composite is beneficial.
Response 2: For better visualization of the distribution of BT nanoparticles in the chitosan matrix, a 50% sharpen correction was applied to the SEM images. In this way, the "incipient dispersion" in the case of 2.5% BT can also be observed. The SEM images used to highlight these details had a magnification of 25,000X. Attempts were made to increase the magnification beyond this value, but the resulting images were blurry and unclear, as shown in the example below. This is the maximum capability of the Scanning Electron Microscope available at our institute. Additionally, arrows were used to denote the location of the nanoparticles in Figure 4.
3. Authors should confirm the interfacial bonding between BT and chitosan by using IR or XPS (better option).
Response 3: FTIR studies of the samples were undertaken to elucidate the interactions between the matrix and BT filler, as seen in the paragraph 3.4 “FTIR characterization” section of the paper.
4. The impact of water absorption on dielectric properties is underexplored. Chitosan’s hygroscopicity may affect long-term stability and should be addressed.
Response 4: To eliminate the effect of water on the dielectric properties all the samples were dried before measurements at 70°C/2h before the measurements and kept in a desiccator. Further the electrical characteristics were determined during 3 different sessions: preliminary characteristics for determining the experimental parameters; low field characterization and high field experiments and results were stable. However, in the aim of the present paper was not to investigate the long-term stability of the dielectric properties, but the suggestion is very good, and we will plan other study (new samples and controlled experimental conditions).
5. In the Conclusion part, the authors should emphasize the relation between dielectric performance and sustainability to highlight the novelty of the work.
Response 5: Conclusion part was revised to highlight the relation between dielectric performance and sustainability, emphasizing the novelty of this work.